# Adaptive Immunity to Viruses: What Did We Learn from SARS-CoV-2 Infection?

**DOI:** 10.3390/ijms232213951

**Published:** 2022-11-12

**Authors:** István Vályi-Nagy, Ferenc Uher, Éva Rákóczi, Zoltán Szekanecz

**Affiliations:** 1South-Pest Hospital Centre, National Institute for Infectious Diseases and Haematology, 1097 Budapest, Hungary; 2Department of Rheumatology, Faculty of Medicine, University of Debrecen, 4032 Debrecen, Hungary

**Keywords:** viral infection, SARS-CoV-2, COVID-19, adaptive immunity, antibodies, B- and T-cell memory, cytokines, cytokine storm

## Abstract

The SARS-CoV-2 virus causes various conditions, from asymptomatic infection to the fatal coronavirus disease 2019 (COVID-19). An intact immune system can overcome SARS-CoV-2 and other viral infections. Defective natural, mainly interferon I- and III-dependent, responses may lead to the spread of the virus to multiple organs. Adaptive B- and T-cell responses, including memory, highly influence the severity and outcome of COVID-19. With respect to B-cell immunity, germinal centre formation is delayed or even absent in the most severe cases. Extrafollicular low-affinity anti-SARS-CoV-2 antibody production will occur instead of specific, high-affinity antibodies. Helper and CD8+ cytotoxic T-cells become hyperactivated and then exhausted, leading to ineffective viral clearance from the body. The dysregulation of neutrophils and monocytes/macrophages, as well as lymphocyte hyperreactivity, might lead to the robust production of inflammatory mediators, also known as cytokine storm. Eventually, the disruption of this complex network of immune cells and mediators leads to severe, sometimes fatal COVID-19 or another viral disease.

## 1. Introduction

The Severe Acute Respiratory Syndrome CoronaVirus-2 (SARS-CoV-2) causes the coronavirus disease 2019 (COVID-19) [1]. SARS-CoV-2 is very similar to other single chain RNA viruses, including SARS-CoV-1 and MERS-CoV, which previously caused SARS and Middle East Respiratory Syndrome (MERS), respectively. The nucleotide homology of SARS-CoV-2 with SARS-CoV-1 and MERS is 80% and 50%, respectively [2]. The virus primarily infects epithelial cells, especially type 2 alveolar epithelial cells by binding to the cell surface angiotensin-converting enzyme 2 (ACE2) receptor. ACE2 is expressed in other cells, including renal, esophageal, gastric, and gut epithelium, as well as myocardium and vascular endothelium and immune cells, including monocyte/macrophages [3,4].

It has become clear that, although virus persistence may also exist in infected organs and tissues, especially in immunocompromised patients [5,6], it is the dysfunction of the immune system and not the virus itself that is mainly responsible for the severe, even fatal outcome of COVID-19 [7,8]. Upon SARS-CoV-2 infection, certain elements of the defence system are not or only moderately activated with significant delay. This results in the disruption of the orchestration of the immune response against the virus. Moreover, other constituents of the immune system become hyperactivated, leading to a cytokine storm and multisystem inflammatory syndrome (MIS). These events lead to the injury of various organs, primarily the lungs [3,7,8,9].

In this review, we will summarize the major features of adaptive immune responses to SARS-CoV-2 infection. We will discuss the specific points of immune system malfunctioning and hyperactivation mentioned above. We will also present some examples of viruses other than SARS-CoV-2 for comparison. We will not discuss innate immunity, sustained immune responses involved in the development of long COVID-19, and responses induced by different vaccines.

## 2. The First Events after SARS-CoV-2 Infection

In brief, the SARS-CoV-2 virus, similar to other SARS-CoV viruses, binds to the angiotensin-converting enzyme 2 (ACE2) surface receptor expressed on alveolar cells and other cells through its spike (S) protein. Binding is promoted by the glycosylation of ACE2, so the sialic acid and ganglioside residues of ACE2 are also involved in virus–epithelial cell binding [10,11].

In general, the virus entering the cell triggers innate immune responses, which will not be discussed in full detail. Briefly, in the cytoplasm of the infected cell, the RNA genome is released from the virus, which makes room for viral replication. Innate immunity to SARS-CoV-2 involves the production of type I and III interferons (IFN), numerous IFN-regulating genes. Type I IFNs are synthesized by all nucleated cells, while type III IFNs are produced mainly by epithelial cells. IFN release occurs rapidly after infection. IFNs inhibit the replication of the virus, and, on the other hand, they transmit the “emergency situation” signal to the surrounding cells, preparing them to defend themselves against the virus. The viral RNA also activates several signalling pathways. Toll-like receptors (TLR) activate the downstream IFN pathways described above. TLRs located in the membrane of endosomes are the first to recognize the foreign nucleic acid molecule. Both TLR7 and TLR3 bind viral RNA. TLR7 and TLR3 activate genes coding for type I and III IFNs via interferon-regulating transcription factor 7 (IRF7) and IRF 3, respectively. These and other signalling pathways are able to initiate the expression of many inflammatory cytokine and chemokine genes leading to systemic inflammation. Collectively, these innate immune processes initiate the mobilization and activation of immune cells, including those participating in the adaptive immune response [8,12,13,14,15].

The orchestration of innate and adaptive immune responses might highly influence the severity and outcome of COVID-19. In the majority of cases, the IFN-I/III response described above occurs very quickly or only with a minimal delay after viral infection. Thus, the infected individual remains symptom-free or, at most, exerts relatively mild symptoms. A well-coordinated innate and adaptive immune response fights SARS-CoV-2 infection in a timely and effective manner. If, on the other hand, the IFN-I/III response is delayed or inadequate, the defence system can no longer control the early replication and spread of the virus in the body. Since the effective T- and B-cell adaptive response is also delayed, the innate immune system tries to compensate by pathological hyperactivity characterized by the appearance of immature myeloid cells and the overproduction of cytokines and chemokines. All of this leads to MIS (cytokine storm) and subsequent multiorgan damage, which requires hospital and sometimes even intensive care treatment and often leads to fatal outcome. The situation becomes critical when the adaptive immune system is unable to respond effectively to the infection even after a long period of time. In this situation, T-cell lymphopenia develops and the production of anti-SARS-CoV-2 antibodies often becomes extreme. The hyperactivity of the innate immune system does not attenuate. This severe condition might occur primarily in old age, when the pool of naive T-cells is already greatly reduced and therefore the proliferation of virus-specific effector lymphocytes requires an even longer time (Figure 1) [3,8,12,13,14,15,16].

## 3. Characteristics of Adaptive Immunity in COVID-19

### 3.1. B-Cell and Antibody Responses

In most infected individuals, seroconversion occurs within 5–15 days. IgM and IgA isotype antibody production occurs first. The IgG response is delayed by 2–3 days and is mainly dominated by antibodies belonging to the IgG1 and IgG3 subclasses. The serum IgM level peaks within 1–2 weeks and then quickly falls below the detection threshold in 2–3 weeks. IgA production reaches its maximum on days 16–22 after infection and disappears from the serum more slowly than IgM. The peak of the IgG response occurs usually between the 3rd and 7th weeks, followed by a plateau and, finally, by a marked serum level decrease. In convalescent patients, anti-viral IgG persists at this lower level from 8–10 weeks to as long as 10–12 months, possibly depending on COVID-19 severity. Somewhat paradoxically, the highest antibody levels are observed in the most severe, often critically ill patients [8,17,18].

The serum levels of virus-neutralizing antibodies usually change parallel with those of IgG isotype anti-spike (anti-S) antibodies. Most neutralizing antibodies recognize the receptor-binding domain (RBD); however, there are also neutralizing antibodies to the N protein N-terminal domain (NTD) domain and the S2 subunit. The N protein is expressed on the surface of infected cells and also occurs in soluble form in the blood. The main, but not exclusive, mechanism of neutralization is that the antibody molecule bound to the appropriate antigenic determinants of the RBD domain inhibits the RBD-ACE2 receptor interaction [8,10,11]. However, the mechanism of in vivo neutralizing activity is much more complex. First, not only RBD-specific antibodies can have such activity. In addition, the neutralizing ability of a given antibody depends on its isotype: which IgG subclass it belongs to and how glycosylated is the Fc part [14,19]. In animal studies, the chemical modification of the Fc part of neutralizing IgG was associated with the loss or at least the impairment of the function of the neutralizing antibody molecule [20]. Moreover, the afucosylated IgG molecules that often appear in COVID-19 are particularly harmful because they bind to the FcγRIIIA (CD16a) molecules on the surface of mast cells, macrophages, and NK cells, leading to the release of inflammatory mediators from CD16a+ cells [14,19].

The answer to the question, why the greatest amount of anti-SARS-CoV-2 antibodies, including significant amounts of neutralizing antibodies, are produced in the most severe COVID-19 patients, lies in the quality of the antibodies and that of the virus-specific B-cell response. A naïve B-cell, if its antigen-recognition receptor specifically binds an antigenic determinant of the virus, can have two types of fate. Further maturation and differentiation can occur extrafollicularly or within the germinal centres of lymphatic organs [8,21]. Similar to other viruses, such as influenza, SARS-CoV-2 first induces an extrafollicular B-cell response followed by a germinal centre response after a few days. During the extrafollicular response, plasmablasts, short-lived plasma cells and atypical memory B-cells with CD21^lo^/CD27−/CD10− phenotype are generated from activated, naïve B lymphocytes. In contrast, during the germinal centre phase, classic CD21+/CD27+/CD10− memory B-cells, as well as long-lived plasma cells are generated [8,21]. A class switch occurs during both phases; however, significant affinity maturation is associated with follicular differentiation only. As a result, plasmablasts and plasma cells developed extrafollicularly secrete, relatively low-affinity, and often polyreactive less-specific antibodies. On the other hand, long-lived plasma cells are characterized by the production of more specific, high-affinity antibodies. In patients with severe COVID-19, the onset of the germinal centre pathway occurs only with a delay or not at all. In these patients, an expansion of atypical memory B-cells can be observed, while the number and proportion of classic memory B-cells decreases. This is most pronounced in cases with fatal outcome where post-mortem spleen and lymph node histology indicates no germinal centres at all. The extrafollicular mechanism tries to compensate for the late or delayed germinal centre response with increased activity. This explains why the highest antibody levels are usually measured in the most severe patients. Yet, even these amounts of antibodies are unable to help to overcome the disease (Figure 2) [8,21,22].

The type of B-cell response, whether dominantly extrafollicular or follicular, is primarily determined by the current microenvironment of the cells, including T helper (T_H_) cells and cytokines. The formation of germinal centres strongly depends on the amount and activity of virus-specific follicular helper T-cells (T_FH_). Most pro-inflammatory cytokines, including interleukin 12 (IL-12), IL-2, IFN-γ, and tumour necrosis factor α (TNF-α), inhibit T_FH_ cell differentiation and germinal centre formation. IL-6, the key cytokine in cytokine storm/MIS associated with severe COVID-19, has the opposite effect. In this respect, IL-6 has a dual role as it promotes germinal centre formation and virus clearance but also promotes systemic inflammation in the early viral and late inflammatory phases of COVID-19, respectively. Therefore, the actual ratio of the various cytokines decides the nature of B-cell response. Lasting humoral protection, namely long-term SARS-CoV-2-specific B-cell memory is provided by classic memory B-cells only, while the lifespan and exact function of atypical memory B-cells is uncertain. Yet, the appearance of atypical memory B-cells has been associated with various chronic viral infections, including HIV and hepatitis B and C [23,24].

The above B-cell-dependent features of COVID-19 may, at least in part, explain autoimmunity observed after SARS-CoV-2 infection [25,26]. The long-lasting extrafollicular response seen in more severe COVID-19 is often associated with the production of autoantibodies. Such autoantibodies include rheumatoid factor, antiphospholipid, anti-myositis-associated 5 (MDA-5), anti-neutrophil cytoplasm (ANCA), anti-β2 microglobulin autoantibodies, and others [25,26,27]. Full-blown autoimmune diseases relatively rarely develop. However, cases of Guillain–Barré syndrome, Miller Fisher syndrome, antiphospholipid syndrome, immune thrombocytopenic purpura, and systemic lupus erythematosus (SLE) have been reported following SARS-CoV-2 infection. It is still not clear whether these are new onset autoimmune phenomena triggered by SARS-CoV-2 or whether the virus only ignites subclinical autoreactive processes leading to the clinical manifestation of the disease [25,26,27]. In any case, many authors suggest connections between more severe COVID-19, autoimmunity, tissue damage, and even long COVID-19 syndrome [25,26,27,28].

Since the SARS-CoV-2 virus that can be transmitted through the respiratory tract, gut, and other epithelial structures, we must briefly discuss the role of locally produced secretory IgA (sIgA) molecules. Secretory IgA-producing plasma cells are located in the upper airways in the subepithelial layer of the mucosa and in the surrounding lymph nodes. In the nose, trachea and bronchi, the virus enters an environment dominated by sIgA dimers. The quantity of sIgA reaches the adult level at the age of 6–12 months. Its virus-neutralizing activity is about 15 times that of the monomeric IgA and IgG molecules found in the serum. Secretory IgA can be detected in the tear and saliva for 49–73 days after infection. Nasopharyngeal-associated (NALT) and bronchial-associated lymphoid tissue (BALT) are more developed in children than in adults. This may partly explain the relative resistance of children and adolescents to COVID-19 compared to adults and elderly people [29,30].

### 3.2. T-Cell Responses

T-cell-mediated immune responses can be observed in almost all SARS-CoV-2-infected individuals. While virus-specific antibodies discussed above are primarily able to inhibit the establishment and early spread of the virus in the body, only the cell-mediated immune response is capable of mitigating the progression of the disease, removing potentially infected cells of vital organs and thus preventing the development of a serious, actually fatal disease [8,31]. In the absence of an adequate T-cell response, for example in untreated HIV-infected or immunocompromised patients, the virus can persist in the tissues for months. The importance of the T-cell responses is also supported by the fact that many patients with agammaglobulinemia or those receiving B-cell-depleting therapies can still recover from COVID-19 [8,31].

In favourable cases, virus-specific CD4+ T-cells can be detected in the blood as early as 2–4 days after the onset of symptoms. Rapid CD4 response has been associated with milder disease course, probably due to increased virus clearance. If the CD4+ T-cell response is delayed, in some cases even for more than 22 days, then the outcome of COVID-19 will be more serious, even fatal. Naïve CD4+ T-cells are able to differentiate into cells that provide various helper and effector functions. As discussed above, Tfh cells promote the differentiation of B lymphocytes into antibody-producing plasma cells and memory B-cells. Th1 cells, in addition to cytokines and chemokines, primarily release IFN-γ, TNF-α, and IL-2. These cells exert indirect antiviral activity. Their main task is to help the division, clonal expansion, and maturation of CD8+ cytotoxic T- cells (T_C_). After viral infection, virus-specific CD8+ T lymphocytes, similar to CD4+ T-cells, rapidly appear in the circulation. Although their quantities are smaller than those of CD4+ T-cells, their appearance and activity usually mean a good prognosis. Their function, almost exclusively, is the destruction of virus-infected human cells and thus the complete removal of the virus from the body. CD8+ T-cells express proteins involved in cell destruction, such as IFN-γ, granzyme B, perforin, and lysosome-associated membrane protein 1 (LAMP-1). In the acute phase of COVID-19, depending on the severity of the disease, T-cells become hyperactive promoting cytokine storm and MIS. They also express various activation markers, including Ki-67, CD38, HLA-DR, and CD69) and the expression of proteins involved in cytotoxicity further increases [3,8,31].

The absolute number of circulating T-cells decreases in parallel with the worsening of the disease. Lymphopenia is one of the biomarkers of MIS [3,32]. Yet, the remaining T lymphocytes usually retain their hyperactive phenotype. The rate of T-cell death can be up to 80% in the circulation, while the proportion of hyperactive T-cells would be around 20%. In case of a successful immune response, 50–60% of SARS-CoV-2-specific CD4+ T-cells become central memory (T_CM_), and 25–40% become effector memory (T_EM_) cells. T_EM_ cells are formed from 40–60% of CD8+ T-cells and then these cells are replaced by other memory T-cell subsets [33,34].

If the immune response is not effective, the virus is not eliminated, COVID-19 stays chronic and the hyperactivated T-cells become exhausted and incapable of functioning. The risk of T-cell exhaustion is particularly high in old age and in certain chronic diseases. During aging, a significant reorganization occurs in the immune system. The proportion of naïve T-cells in the body decreases, and thus fewer clones will be able to recognize a new pathogen. In parallel to the narrowing of the T lymphocyte repertoire, the ability of individual T-cell clones to expand and differentiate also decreases. Fewer T_FH_ cells are generated and the cytotoxic activity of CD8+ T_C_-cells also becomes attenuated. One of the reasons for the relative resistance of children to COVID-19 is that they have a higher proportion of naïve T-cells; thus, a more extensive T-cell repertoire is available upon infection. The amount and functional capability of regulatory T-cells (T_REG_) also decreases with aging. This also favours the development of an unbalanced immune response with severely decreased efficiency resulting in cytokine storm and MIS [31,35]. Some studies suggest that it is not primarily the insufficient anti-SARS-CoV-2 immunity but rather hyperreactivity that is associated with the development of critical COVID-19 [16].

Both CD4+ and CD8+ memory T-cells can be detected in the blood of patients who recovered from COVID-19. The quantity of these cells stabilizes in the circulation with a half-life of about 200 days. These cells, regardless of the severity of COVID-19, exert a phenotype (TCF-1+, CD45RA+, CD95+) characteristic of stem cell-like memory T-cells (T_SCM_). In general, these cells recognize the same antigenic determinants as T-cells in the acute phase of the disease. The T_SCM_ phenotype is characteristic of T-cells in the early differentiation state. In case of a new antigenic stimulus, different effector T-cells, including T_FH_ and T_C_ cells, can be generated from T_SCM_ cells, providing much faster and more efficient humoral and cellular immune responses than before. Anti-SARS-CoV-2-specific memory T-cells can be detected in about 93% of asymptomatic patients, while seropositivity can be detected in only 60% or less. Thus, an adaptive, T-cell-mediated immune response and memory can also develop in asymptomatic infected individuals [8,14,31,34].

It is important that T lymphocytes can only recognize properly processed and presented antigenic determinants. The proteins of the SARS-CoV-2 virus contain nearly 1400 antigenic determinants that can be recognized by T lymphocytes after antigen processing. The real number can be much lower than this as the T-cells of an individual with a given HLA haplotype can only recognize 30–40 immunodominant viral determinants. No more than a third of them can be found in the S protein. Most T-cell determinants can be linked to other structural proteins, such as the N and M, as well as nsp and other accessory proteins. Thus, the anti-SARS-CoV-2 T-cell response, in contrast to the B-cell or antibody response, is not limited to the S protein, which is mostly affected by mutations. That is why numerous T-cell clones induced by previous virus variants, unlike antibodies, are also able to recognize newer variants of SARS-CoV-2. About 75–85% of the T-cell determinants are retained even in the most highly mutated B.1.1.529 (omicron) variant. On the other hand, the convalescent plasma obtained from patients who recovered from COVID-19 during earlier pandemic waves does not or only minimally reacts with the S protein of the omicron variant [31,34,36,37].

It is still a debate whether activated and hyperactivated T lymphocytes are involved in the development of severe COVID-19 or not. In principle, T-cell activation itself can lead to tissue damage. However, such cells have not been detected in the blood, lungs, or other vital organs of patients with severe COVID-19 so far, although this does not rule out their existence. As discussed above, low or even very low T-cell counts and severe lymphopenia has been detected in patients requiring intensive treatment [3,32]. It is most likely that hyperactive T lymphocytes primarily contribute to the increased function of myeloid cells through their cytokine production, so they indirectly participate in the development of cytokine storm and MIS [3,8,16,32].

It has been questionable for some time as to whether non-conventional, non-MHC-restricted T-cells play any role during COVID-19 or not. However, it has now become clear that the amount of mucosa-associated, invariant T-cells (MAIT, MR1-restricted mucosal-associated invariant T-cells), CD1d-restricted natural killer cells (iNKT, CD1d-restricted invariant natural killer cells), and gamma-delta T-cells (γδT-cells, gammadelta T-cells) decrease significantly in COVID-19. These cells together account for about 10% of circulating T lymphocytes. The small amount of non-conventional T-cells enter an activated state, which is indicated by their high expression of IL-17A and CD69 and the diminished production of IFN-γ. These cells also appear in bronchoalveolar lavage fluid (BALF) where, in addition to IL-17A, other pro-inflammatory cytokines, including IL-1β, IL-6, IL-8, IL-15, IL-18, and IFN-α, become secreted. The migration of these non-conventional T-cells to the lung is induced by CCL20, CXCL10, CCL11, and CXCL16 chemokines released from activated myeloid cells. It is unlikely that the virus directly activates these cells. It is the inflammatory environment or secondary bacterial infection that may rather be responsible for the enhanced function of these cells. Moreover, the amount of non-conventional T-cells in the body decreases with age. Over the age of 70–80, these cells can almost completely disappear from the blood and lymphatic organs. Further studies are needed in order to clarify the role these T-cells in COVID-19 [8,38,39].

### 3.3. Immunological Memory in COVID-19

The function of the immunological memory is to protect the body from a possible reinfection, or at least from the disease caused by the repeated infection. It has four main components: antibodies (long-lived plasma cells) and memory B-, as well as CD4+ and CD8+ memory T-cells. The role of these four players in COVID-19 was in part discussed above [40,41]. With respect to the type of memory associated with SARS-CoV-2 infection, in a large study, the evolution of the amount of circulating anti-RBD IgG and anti-S IgA antibodies, RBD-specific memory B-cells, as well as virus-specific CD4+ and CD8+ memory T-cells was monitored in 188 convalescent patients for eight months [42]. Already one month after the onset of symptoms, the majority of patients were positive for all five measured parameters, and five months later, 95% of them were still positive for at least three parameters. In 92% of subjects, the number of CD4+ memory T-cells did not decrease even within 8 months, while the number of CD8+ memory T-cells decreased to half by the end of the 8th month in comparison to baseline. In general, the frequency of RBD-specific memory B-cells increased over time. Importantly, there were enormous individual differences among patients regarding the type of memory that developed. It was largely independent of the severity of preceding COVID-19, whether a given individual was positive for three, four, or all five parameters [42]. Antibody positivity was the most incidental; therefore, simple antibody testing itself is not suitable to assess immunological memory in COVID-19 [40,41,42].

The ultimate question is how durable the specific memory is. In the case of SARS-CoV-2, due to the relatively short time since the outbreak of the epidemic, we cannot yet determine this for sure. A drop in antibody levels after infection is normal. After the SARS-CoV-1 epidemic in China in 2003–2004, specific antibodies and T-cells were detected in the blood of convalescent patients even after 4 and 17 years, respectively [43,44,45]. In a recent study, investigators followed people who had recovered from COVID-19. The antibody levels in these individuals decreased quickly for the first 2–3 months after infection but then the curve started to flatten. Many recovered patients had detectable anti-spike IgM, IgG, and neutralizing antibody levels for 8 months, sometimes up to 450 days [46,47]. These patients also had long-term T-cell responses [46,47]. It has been postulated, based on encouraging data, that pre-existing immune reactivity to SARS-CoV-2 may exist to some degree in the general population before the actual infection, It is hypothesized that this might be due to immunity from previously acquired common cold infections [45].

On the other hand, SARS-CoV-2, as well as endemic coronaviruses, might re-infect those who have previously had the disease [8,48]. In addition, those COVID-19 patients who have experienced an endemic coronavirus infection in the previous 3 years are significantly less likely to require intensive care than other patients [49]. One possible explanation for this is that T-cells specific for these endemic coronaviruses, which cross-react with SARS-CoV-2, can be detected in a significant proportion of individuals. For example, CD4+ memory T-cells are present in the blood of 70–80% of the population, while CD8+ T-cells are rarer, occurring in only 10–20% of people. The frequency of such cross-reacting T-cells is the highest in children and the young and decreases to almost zero with aging. Thus, some degree of T-cell memory might exist even for years in patients who have recovered from COVID-19. However, the extent and time course of this T-cell memory response can be very different among individuals [33,46,47,49,50].

T-cell responses and memory are also relevant for antiviral responses against different SARS-CoV-2 variants. With respect to humoral immunity, attenuated antibody responses were detected against the omicron compared to the delta and other previous variants [51]. On the other hand, similar CD4+ and CD8+ T-cell responses were observed against the omicron, delta, and other variants in convalescent individuals. This indicates that ancestral SARS-CoV-2-driven T-cells cross-react with the omicron variant [37,52].

One of the most important questions is of course: if there is memory in convalescent patients, then what is it enough for? Does it protect from reinfection with SARS-CoV-2 or at least from more severe illness? What is certain is that, usually after the appearance of different virus variants, reinfection rarely occurs. The second and maybe consecutive infections are usually mild or even asymptomatic. This definitely points to the existence of a longer-term memory, but it tells us little about its true effects and especially its duration. It might take years to gather a reliable amount of data on this [33,49,50].

## 4. Cytokine Storm and MIS

COVID-19 comprises three stages [53,54,55]. Stage 1 is the period of early viral infection with fever, respiratory or gastrointestinal symptoms, and lymphopenia. Stage 2 is the pulmonary phase with pneumonia. Finally, Stage 3 is the phase of MIS, occasionally accompanied by cytokine storm [53,54,55]. This late stage of COVID-19 also involves, among other mechanisms, bradykinin storm [56], the activation of coagulation and complement cascades [57], endothelitis, vascular leak and oedema [57], microthrombotic events [58], and the formation of neutrophil extracellular traps (NET) [59]. The presence of cytokine storm/MIS should be confirmed by clinical, imaging, and laboratory markers [32,53,60,61]. Laboratory biomarkers, such as C-reactive protein (CRP), ferritin, D-dimer, cardiac troponin (cTn), NT-proBNP, lymphopenia, neutropenia, and, if available, circulating IL-6 levels have been associated with MIS in Stages 2–3 and also with poor outcomes of COVID-19 [32,61].

Cytokine storm/MIS usually develops in the second week after the first symptoms of COVID-19. The activated neutrophils, monocytes/macrophages, and NK cells of affected patients, as well as dying epithelial and endothelial cells exposed to the direct cytopathic effect of the virus, secrete numerous cytokines, chemokines, and other inflammatory mediators both locally and systemically. The resulting inflammatory environment naturally affects the adaptive immune system. Activated T-cells further stimulate the process through positive feedback mechanisms and also through the mediation of cytokines. The combination of these mechanisms triggers an, in severe cases almost uncontrollable, inflammatory “spiral”, which can fatally damage various vital organs. In the most affected lung, for example, the damage, including cell death, oedema, and vascular injury, might lead to the development of acute respiratory distress syndrome (ARDS). Later multi-organ failure might also develop. During a cytokine storm, among cytokines and growth factors, IL-1β, IL-6, IL-7, IL-8, IL-9, IL-10, IL-17, TNF-α, IFN-γ, granulocyte (G-CSF), and granulocyte-macrophage colony-stimulating factors (GM-CSF), as well as vascular endothelial growth factor (VEGF), are released in great quantities. This is accompanied by the abundant production of chemokines, including CCL2/MCP-1, CCL3/MIP-1α, CCL5/RANTES, CCL8/MCP2, CXCL2/SDF-1, CXCL4/PF4, CXCL8/IL-8, CXCL9/MIG, CXCL10/IP-10, and CXCL16. This hyper-inflammatory environment (MIS), primarily as a result of high TNF-α and IFN-γ levels, can cause pyroptosis, apoptosis, and necrosis (PANoptosis) in almost any cell, causing further tissue and multiorgan damage. For example, it also affects the cells of the adaptive immune system, which is why lymphopenia develops in severe COVID-19 patients. In addition, during SARS-CoV-2 infection also triggers autoinflammation, which is characterized by the activation of the NLRP3 (NOD-, LRR-, and pyrin domain-containing protein 3) inflammasome, an intracellular virus sensor complex. NLRP3 is activated in monocyte/macrophages, which can cleave through the caspase-1 protein pro-IL-1β and pro-IL-18 molecules. As for virus-infected endothelium, the expression ICAM-1 and VCAM-1 increases on the surface of activated endothelial cells. This increases the risk of developing microthrombi. Activated platelets release additional inflammatory mediators, including histamine, IL-1β, CCL3/MIP-1α, CXCL4/PF4, prostaglandins, and thromboxane A2 [53,54,55]. In addition, insufficiently efficient or delayed type I and III IFN responses also significantly contribute to the development of cytokine storm/MIS. Thus, the IFN-I/III response is a double-edged sword. In the early stages of COVID-19, IFNs play an essential role in protection against the virus. However, after a few days, during which period of time the virus can unlimitedly proliferate in the body, the IFN responses increase the cytopathic effect of the virus and enhance the inflammatory process (Figure 3) [62].

Cytokine storm/MIS may differ in COVID-19 and other viral infections. Higher IL-6 but lower IL-1α levels have been detected in patients with primary COVID-19 compared to influenza [63]. It seems that only 5–11% of severe COVID-19 patients have strong cytokine storm, which is less frequent than seen in influenza [63]. Some authors even questioned the existence of a cytokine storm in COVID-19 as the majority of patients with severe COVID-19 had cytokine levels comparable with trauma or cardiac arrest, conditions that are not noted for a cytokine storm [64]. However, plasma IL-1β, IL-6, IL-8, and chemokine levels were associated with intensive care unit admission and the death of severe COVID-19 patients. The adjusted odds ratios of these mediators varied between 5 and 10 [63]. The detection of cytokine storm/MIS also exerts a time window of 7 to 14 days after the first symptoms. Systemic inflammation might not be present in the early viral stages or in the late phase with multiorgan damage [3,65].

## 5. Conclusions

If the innate and adaptive immune systems work in a balanced and coordinated manner, the human body is able to overcome the SARS-CoV-2 infection. If any element of the defence system is not able to fully perform its task, is not activated at the right time, or simply does not work properly, then the harmony within the system ceases and this can have serious pathological consequences. If the IFN I/III responses are disturbed, the virus can multiply almost indefinitely in the respiratory tract. The dysregulation of myeloid phagocytic cells causes an increase in pro-inflammatory cytokines and chemokines, leading to the hyperreactivity of lymphocytes, the further abundant production of inflammatory mediators (cytokine storm), and eventually MIS. Hyperactivated T-cells become partially paralyzed or partially die (lymphopenia). If the number and/or activity of T_FH_ cells is not adequate, the formation of germinal centres is delayed or even absent in the most severe patients. In such cases, antibody production occurs permanently extrafollicularly and primarily poly- and often autoreactive and low-affinity anti-SARS-CoV-2 antibodies are formed. On the other hand, the B-cells that undergo affinity maturation in the germinal centre and the plasma cells generated from them, which would be capable of secreting more specific, high-affinity antibodies, are absent. Hyperactivation, exhaustion, and then the partial death of Th_1_ and especially CD8+ T_C_ cells, which otherwise play a key role in eradicating the already extensive infection hinders the complete removal of the virus from the body. Since the immune system works as an extremely finely regulated, complex network, the malfunctions discussed above never occur in an isolated manner but reinforce each other. This is what makes the SARS-CoV-2 infection severe, sometimes fatal.

## Figures and Tables

**Figure 1 ijms-23-13951-f001:**
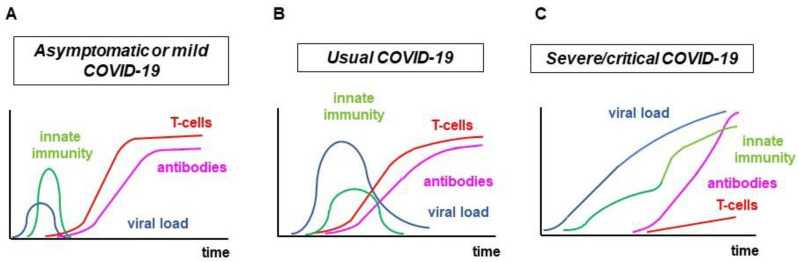
Immunological processes in COVID-19 with different severity. (**A**). asymptomatic or mild COVID-19; (**B**). moderate/usual COVID-19; (**C**). severe/critical COVID-19.

**Figure 2 ijms-23-13951-f002:**
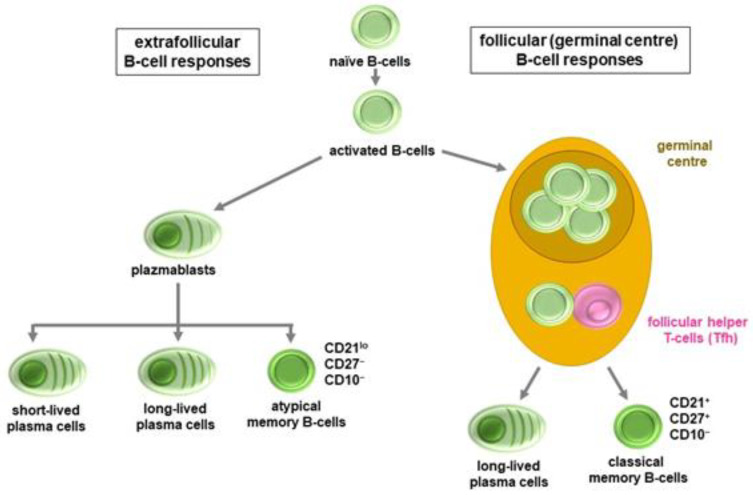
Extrafollicular and follicular B-cell responses.

**Figure 3 ijms-23-13951-f003:**
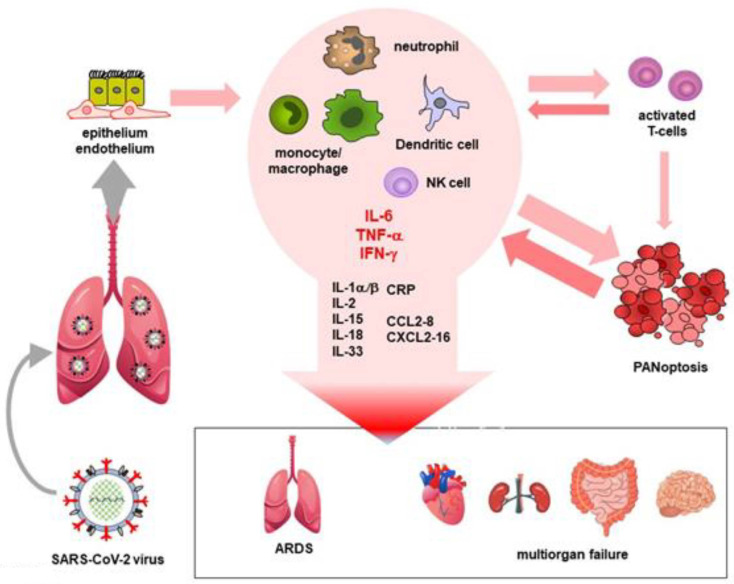
Development of cytokine storm and multiorgan involvement in COVID-19.

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
