# Peer review of "Adaptive Immunity to Viruses: What Did We Learn from SARS-CoV-2 Infection?"

_ijms, 2022, doi:10.3390/ijms232213951_

Round 1
Reviewer 1 Report
No comment
Author Response
Dear Reviewer, many thanks for the very positive assessment and review. Zoltan Szekanecz
Reviewer 2 Report
The review manuscript summarizes the adaptive immunity to SARS CoV-2 infection. The intact immune system plays a critical role in eliminating SARS-CoV2 and other viruses, and controlling the infections. The defect of any immune parts may cause uncontrolled SARS Cov-2 infection and COVID-19 symptoms, which can be very severe and even to death. This review focused on the adaptive immunity of SARS-CoV-2, including B cell immunity, T cell immunity, B and T immune memory, and cytokine storm and multi-organ inflammatory symptom (MIS). The information of this review provides valuable understanding of SARS-CoV-2 infection and the immune-pathogenesis of COVID-19.
The manuscript is generally well written and clear, there are some minor concerns as following:
1, Line 50, the similarly to should be similar to.
2, Lines 58-59, The IFN and numerous IFN regulating genes seems here parallel to TLRs and other signaling pathways. TLRs and other signaling pathways activate the downstream IFN and related gene expressions. Please rewrite accordingly.
3, Lines 65-66, TLR7, specific for single-stranded RNA (ssRNA), but also TLR3 can also bind viral RNA. The sentence is not correct in grammar, please rewrite.
4, Line 70, In parallel should be Collectively.
5, Line 81, what does the natural immune system mean here? To keep consistent, the natural immune/immunity should be innate immune/immunity.
6, Line 105, what is the meaning of severity V-2 IgG here? Please correct.
7, Lines 110-111, N, S2 and nucleocapsid protein, is it correct? What is nucleocapsid protein and is it different form N protein?
8, Lines 157-158, The IL-6 effect here seems paradoxical, it is a significant contributor to inflammation, but here it helps germinal centre formation, promoting virus clearance. How to reconcile these two different effects?
9, Lines 242-243, “that is not primarily---” should be “that it is not primarily---“. The “dev elopement” should be “development”.
10, Line 306, eight months should be for or in eight months?

Author Response
Dear Reviewer, many thanks for the very positive assessment and review. We corrected all errors listed. Many thanks for your great work.
Here are the responses and corrections:
1, Line 50, the similarly to should be similar to. - now corrected
2, Lines 58-59, The IFN and numerous IFN regulating genes seems here parallel to TLRs and other signaling pathways. TLRs and other signaling pathways activate the downstream IFN and related gene expressions. Please rewrite accordingly. - yes, we now added info that TLRs are upstream and they regulate IFN responses
3, Lines 65-66, TLR7, specific for single-stranded RNA (ssRNA), but also TLR3 can also bind viral RNA. The sentence is not correct in grammar, please rewrite. - corrected
4, Line 70, In parallel should be Collectively. - corrected
5, Line 81, what does the natural immune system mean here? To keep consistent, the natural immune/immunity should be innate immune/immunity. - we changed natural to innate at every instance
6, Line 105, what is the meaning of severity V-2 IgG here? Please correct. - sorry, V-2 IgG remained there in error, deleted
7, Lines 110-111, N, S2 and nucleocapsid protein, is it correct? What is nucleocapsid protein and is it different form N protein? - yes, N is nucleocapsid, so the word nucleocapsid is now deleted
8, Lines 157-158, The IL-6 effect here seems paradoxical, it is a significant contributor to inflammation, but here it helps germinal centre formation, promoting virus clearance. How to reconcile these two different effects?
- the GC formation and antiviral effects is in the early phase, while the inflammatory effect is in the late phase of COVID-19, now clarified in the paper
9, Lines 242-243, “that is not primarily---” should be “that it is not primarily---“. The “dev elopement” should be “development”. - both corrected
10, Line 306, eight months should be for or in eight months? - corrected
Again, many thanks for the great suggestions
Zoltan Szekanecz
Reviewer 3 Report
In the current review by Vályi-Nagy I et al., authors have nicely summarized the major features and players in the adaptive T cell responses against SARS-CoV-2 infection. The review is we written, easy to follow, and presented in very engaging way.
Authors have focused mainly on adaptive immune regulation and activation part during COVID-19 infection and precisely focused on the main components of regulators of adaptive immune system.
It would be great of they can additionally touch base on convalescent plasma therapy as well as protection against SARS-CoV-2 upon vaccination. Long term effects on immunocompromised patients and healthy individuals getting COVID-19 infection should also be mentioned and how the fine tuning of innate and adaptive immune system could shape the outcome of the disease will give extra edge for the review.
Line 131: Lymphocytic choriomeningitis virus (LCMV) establishes acute or chronic virus infections and the clearance is mainly dependent on activation of T cells and not so much (or very little) by antibody neutralization. ON the other hand, clearance of cytopathic viruses is mainly B cell dependent. You may want to change LCMV to VSV etc. or provide reference for LCMV part.
Minor points:
First line of the abstract is written in too basic way. I would rephrase this line with different words. Also, use of word “may” suggest possibility, however, SARS-CoV-2 causes a lot of asymptomatic conditions.
Lines 108-114: add references
Line 125: SARS-CoV-2

Author Response
Dear Madam/Sir
Many thanks for your great job in reviewing our paper, we now tried to incorpoprate all your remarks and suggestions into the manuscript as follows:
In the current review by Vályi-Nagy I et al., authors have nicely summarized the major features and players in the adaptive T cell responses against SARS-CoV-2 infection. The review is we written, easy to follow, and presented in very engaging way. - many thanks for the positive approach
Authors have focused mainly on adaptive immune regulation and activation part during COVID-19 infection and precisely focused on the main components of regulators of adaptive immune system. - the innate responses are also discussed in part
It would be great of they can additionally touch base on convalescent plasma therapy as well as protection against SARS-CoV-2 upon vaccination. Long term effects on immunocompromised patients and healthy individuals getting COVID-19 infection should also be mentioned and how the fine tuning of innate and adaptive immune system could shape the outcome of the disease will give extra edge for the review.
- we did not want to add convalescent plasma therapy as this is not effective at all in large trials. Also vaccination was out of the scope of this paper and the length limitations also limit adding any more information. Discussing vaccination responses would require a lot more space. We already added a paragraph with some information of fine tuning between innate and adaptive immunity: "The orchestration of innate and adaptive immune responses might highly influence the severity and outcome of COVID-19..." section
Line 131: Lymphocytic choriomeningitis virus (LCMV) establishes acute or chronic virus infections and the clearance is mainly dependent on activation of T cells and not so much (or very little) by antibody neutralization. ON the other hand, clearance of cytopathic viruses is mainly B cell dependent. You may want to change LCMV to VSV etc. or provide reference for LCMV part.
- the reviewer is right, sorry, we deleted the LCMV part
Minor points:
First line of the abstract is written in too basic way. I would rephrase this line with different words. Also, use of word “may” suggest possibility, however, SARS-CoV-2 causes a lot of asymptomatic conditions. - this line is now rephrased and "may" deleted
Lines 108-114: add references - references added
Line 125: SARS-CoV-2 - corrected
again we would like to thank the Reviewer for the valuable comments and remarks
Zoltan Szekanecz